# The Pareto Regret Frontier

**Wouter M. Koolen**
Queensland University of Technology
`wouter.koolen@qut.edu.au`

## Abstract

Performance guarantees for online learning algorithms typically take the form of *regret bounds*, which express that the cumulative loss overhead compared to the best expert in hindsight is small. In the common case of large but structured expert sets we typically wish to keep the regret especially small compared to simple experts, at the cost of modest additional overhead compared to more complex others. We study which such regret trade-offs can be achieved, and how.

We analyse regret w.r.t. each individual expert as a multi-objective criterion in the simple but fundamental case of absolute loss. We characterise the achievable and Pareto optimal trade-offs, and the corresponding optimal strategies for each sample size both exactly for each finite horizon and asymptotically.

## 1 Introduction

One of the central problems studied in online learning is *prediction with expert advice*. In this task a learner is given access to $K$ strategies, customarily referred to as *experts*. He needs to make a sequence of $T$ decisions with the objective of performing as well as the best expert in hindsight. This goal can be achieved with modest overhead, called *regret*. Typical algorithms, e.g. Hedge [1] with learning rate $\eta = \sqrt{8/T \ln K}$, guarantee

$$L_T - L_T^k \leq \sqrt{T/2 \ln K} \qquad \text{for each expert } k. \tag{1}$$

where $L_T$ and $L_T^k$ are the cumulative losses of the learner and expert $k$ after all $T$ rounds.

Here we take a closer look at that right-hand side. For it is not always desirable to have a *uniform* regret bound w.r.t. all experts. Instead, we may want to single out a few special experts and demand to be really close to them, at the cost of increased overhead compared to the rest. When the number of experts $K$ is large or infinite, such favouritism even seems unavoidable for non-trivial regret bounds. The typical proof of the regret bound (1) suggests that the following can be guaranteed as well. For each choice of probability distribution $q$ on experts, there is an algorithm that guarantees

$$L_T - L_T^k \leq \sqrt{T/2(-\ln q(k))} \qquad \text{for each expert } k. \tag{2}$$

However, it is not immediately obvious how this can be achieved. For example, the Hedge learning rate $\eta$ would need to be tuned differently for different experts. We are only aware of a single (complex) algorithm that achieves something along these lines [2]. On the flip side, it is also not obvious that this trade-off profile is optimal.

In this paper we study the Pareto (achievable and non-dominated) regret trade-offs. Let us say that a candidate trade-off $\langle r_1, \dots, r_K \rangle \in \mathbb{R}^K$ is *T-realisable* if there is an algorithm that guarantees

$$L_T - L_T^k \leq r_k \qquad \text{for each expert } k.$$

Which trade-offs are realisable? Among them, which are optimal? And what is the strategy that witnesses these realisable strategies?

## 1.1 This paper

We resolve the preceding questions for the simplest case of absolute loss, where $K = 2$. We first obtain an exact characterisation of the set of realisable trade-offs. We then construct for each realisable profile a witnessing strategy. We also give a randomised procedure for optimal play that extends the randomised procedures for balanced regret profiles from [3] and later [4, 5].

We then focus on the relation between priors and regret bounds, to see if the particular form (2) is achievable, and if so, whether it is optimal. To this end, we characterise the asymptotic Pareto frontier as $T \to \infty$. We find that the form (2) is indeed achievable but fundamentally sub-optimal. This is of philosophical interest as it hints that approaching absolute loss by essentially reducing it to information theory (including Bayesian and Minimum Description Length methods, relative entropy based optimisation (instance of Mirror Descent), Defensive Forecasting etc.) is *lossy*.

Finally, we show that our solution for absolute loss equals that of $K = 2$ experts with bounded linear loss. We then show how to obtain the bound (1) for $K \geq 2$ experts using a recursive combination of two-expert predictors. Counter-intuitively, this cannot be achieved with a *balanced* binary tree of predictors, but requires the most *unbalanced* tree possible. Recursive combination with non-uniform prior weights allows us to obtain (2) (with higher constant) for any prior $q$.

## 1.2 Related work

Our work lies in the intersection of two lines of work, and uses ideas from both. On the one hand there are the game-theoretic (minimax) approaches to prediction with expert advice. In [6] Cesa-Bianchi, Freund, Haussler, Helmbold, Schapire and Warmuth analysed the minimax strategy for absolute loss with a known time horizon $T$. In [5] Cesa-Bianchi and Shamir used random walks to implement it efficiently for $K = 2$ experts or $K \geq 2$ static experts. A similar analysis was given by Koolen in [4] with an application to tracking. In [7] Abernethy, Langford and Warmuth obtained the optimal strategy for absolute loss with experts that issue binary predictions, now controlling the game complexity by imposing a bound on the loss of the best expert. Then in [3] Abernethy, Warmuth and Yellin obtained the worst case optimal algorithm for $K \geq 2$ arbitrary experts. More general budgets were subsequently analysed by Abernethy and Warmuth in [8]. Connections between minimax values and algorithms were studied by Rakhlin, Shamir and Sridharan in [9].

On the other hand there are the approaches that do not treat all experts equally. Freund and Schapire obtain a non-uniform bound for Hedge in [1] using priors, although they leave the tuning problem open. The tuning problem was addressed by Hutter and Poland in [2] using two-stages of Follow the Perturbed Leader. Even-Dar, Kearns, Mansour and Wortman characterise the achievable trade-offs when we desire especially small regret compared to a fixed average of the experts' losses in [10]. Their bounds were subsequently tightened by Kapralov and Panigrahy in [11]. An at least tangentially related problem is to ensure smaller regret when there are several good experts. This was achieved by Chaudhuri, Freund and Hsu in [12], and later refined by Chernov and Vovk in [13].

## 2 Setup

The *absolute loss game* is one of the core decision problems studied in online learning [14]. In it, the learner sequentially predicts $T$ binary outcomes. Each round $t \in \{1, \ldots, T\}$ the learner assigns a probability $p_t \in [0, 1]$ to the next outcome being a 1, after which the actual outcome $x_t \in \{0, 1\}$ is revealed, and the learner suffers *absolute loss* $|p_t - x_t|$. Note that absolute loss equals expected $0/1$ loss, that is, the probability of a mistake if a "hard" prediction in $\{0, 1\}$ is sampled with bias $p$ on 1.

Realising that the learner cannot avoid high cumulative loss without assumptions on the origin of the outcomes, the learner's objective is defined to ensure low cumulative loss compared to a fixed set of baseline strategies. Meeting this goal ensures that the easier the outcome sequence (i.e. for which some reference strategy has low loss), the lower the cumulative loss incurred by the learner.

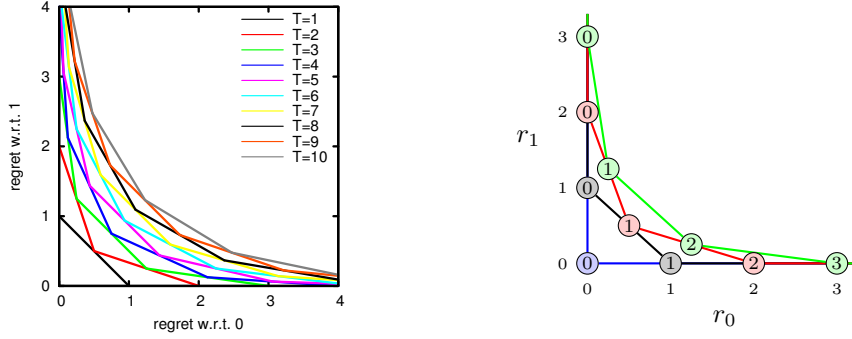

(a) The Pareto trade-off profiles for small $T$. The sets $\mathbb{G}_T$ consist of the points to the north-east of each curve.

(b) Realisable trade-off profiles for $T = 0, 1, 2, 3$. The vertices on the profile for each horizon $T$ are numbered $0, \ldots, T$ from left to right.

Figure 1: Exact regret trade-off profile

The *regret* w.r.t. the strategy $k \in \{0, 1\}$ that always predicts $k$ is given by [1]

$$R_T^k := \sum_{t=1}^{T} \big( |p_t - x_t| - |k - x_t| \big).$$

Minimising regret, defined in this way, is a multi-objective optimisation problem. The classical approach is to "scalarise" it into the single objective $R_T := \max_k R_T^k$, that is, to ensure small regret compared to the best expert in hindsight. In this paper we study the full Pareto trade-off curve.

**Definition 1.** A candidate trade-off $\langle r_0, r_1 \rangle \in \mathbb{R}^2$ is called $T$-*realisable* for the $T$-round absolute loss game if there is a strategy that keeps the regret w.r.t. each $k \in \{0, 1\}$ below $r_k$, i.e. if

$$\exists p_1 \forall x_1 \cdots \exists p_T \forall x_T : R_T^0 \leq r_0 \text{ and } R_T^1 \leq r_1$$

where $p_t \in [0, 1]$ and $x_t \in \{0, 1\}$ in each round $t$. We denote the set of all $T$-realisable pairs by $\mathbb{G}_T$.

This definition extends easily to other losses, many experts, fancy reference combinations of experts (e.g. shifts, drift, mixtures), protocols with side information etc. We consider some of these extension in Section 5, but for now our goal is to keep it as simple as possible.

## 3 The exact regret trade-off profile

In this section we characterise the set $\mathbb{G}_T \subset \mathbb{R}^2$ of $T$-realisable trade-offs. We show that it is a convex polygon, that we subsequently characterise by its vertices and edges. We also exhibit the optimal strategy witnessing each Pareto optimal trade-off and discuss the connection with random walks. We first present some useful observations about $\mathbb{G}_T$.

The linearity of the loss as a function of the prediction already renders $\mathbb{G}_T$ highly regular.

**Lemma 2.** *The set $\mathbb{G}_T$ of $T$-realisable trade-offs is convex for each $T$.*

*Proof.* Take $\boldsymbol{r}^A$ and $\boldsymbol{r}^B$ in $\mathbb{G}_T$. We need to show that $\alpha \boldsymbol{r}^A + (1-\alpha) \boldsymbol{r}^B \in \mathbb{G}_T$ for all $\alpha \in [0, 1]$. Let $A$ and $B$ be strategies witnessing the $T$-realisability of these points. Now consider the strategy that in each round $t$ plays the mixture $\alpha p_t^A + (1 - \alpha) p_t^B$. As the absolute loss is linear in the prediction, this strategy guarantees $L_T = \alpha L_T^A + (1-\alpha) L_T^B \leq L_T^k + \alpha r_k^A + (1-\alpha) r_k^B$ for each $k \in \{0, 1\}$. $\square$

Guarantees violated early cannot be restored later.

**Lemma 3.** *A strategy that guarantees $R_T^k \leq r_k$ must maintain $R_t^k \leq r_k$ for all $0 \leq t \leq T$.*

*Proof.* Suppose toward contradiction that $R_t^k > r_k$ at some $t < T$. An adversary may set all $x_{t+1} \ldots x_T$ to $k$ to fix $L_T^k = L_t^k$. As $L_T \geq L_t$, we have $R_T^k = L_T - L_T^k \geq L_t - L_t^k = R_t^k > r_k$. $\square$

The two extreme trade-offs $\langle 0, T \rangle$ and $\langle T, 0 \rangle$ are Pareto optimal.

**Lemma 4.** *Fix horizon $T$ and $r_1 \in \mathbb{R}$. The candidate profile $\langle 0, r_1 \rangle$ is $T$-realisable iff $r_1 \geq T$.*

*Proof.* The static strategy $p_t = 0$ witnesses $\langle 0, T \rangle \in \mathbb{G}_T$ for every horizon $T$. To ensure $R_T^1 < T$, any strategy will have to play $p_t > 0$ at some time $t \leq T$. But then it cannot maintain $R_t^0 = 0$. $\square$

It is also intuitive that maintaining low regret becomes progressively harder with $T$.

**Lemma 5.** $\mathbb{G}_0 \supset \mathbb{G}_1 \supset \ldots$

*Proof.* Lemma 3 establishes $\supseteq$, whereas Lemma 4 establishes $\neq$. $\square$

We now come to our first main result, the characterisation of $\mathbb{G}_T$. We will directly characterise its south-west frontier, that is, the set of Pareto optimal trade-offs. These frontiers are graphed up to $T = 10$ in Figure 1a. The vertex numbering we introduce below is illustrated by Figure 1b.

**Theorem 6.** *The Pareto frontier of $\mathbb{G}_T$ is the piece-wise linear curve through the $T + 1$ vertices*

$$\langle f_T(i), f_T(T - i) \rangle \quad \text{for } i \in \{0, \ldots, T\} \qquad \text{where} \qquad f_T(i) := \sum_{j=0}^{i} j 2^{j-T} \binom{T - j - 1}{T - i - 1}.$$

*Moreover, for $T > 0$ the optimal strategy at vertex $i$ assigns to the outcome $x = 1$ the probability*

$$p_T(0) := 0, \quad p_T(T) := 1, \quad \text{and} \quad p_T(i) := \frac{f_{T-1}(i) - f_{T-1}(i-1)}{2} \quad \text{for} \quad 0 < i < T,$$

*and the optimal probability interpolates linearly in between consecutive vertices.*

*Proof.* By induction on $T$. We first consider the base case $T = 0$. By Definition 1
$$\mathbb{G}_0 = \{ \langle r_0, r_1 \rangle \mid r_0 \geq 0 \text{ and } r_1 \geq 0 \}$$
is the positive orthant, which has the origin as its single Pareto optimal vertex, and indeed $\langle f_0(0), f_0(0) \rangle = \langle 0, 0 \rangle$. We now turn to $T \geq 1$. Again by Definition 1 $\langle r_0, r_1 \rangle \in \mathbb{G}_T$ if

$$\exists p \in [0, 1] \forall x \in \{0, 1\} : \langle r_0 - |p - x| + |0 - x|, r_1 - |p - x| + |1 - x| \rangle \in \mathbb{G}_{T-1},$$

that is if

$$\exists p \in [0, 1] : \langle r_0 - p, r_1 - p + 1 \rangle \in \mathbb{G}_{T-1} \text{ and } \langle r_0 + p, r_1 + p - 1 \rangle \in \mathbb{G}_{T-1}.$$

By the induction hypothesis we know that the south-west frontier curve for $\mathbb{G}_{T-1}$ is piecewise linear. We will characterise $\mathbb{G}_T$ via its frontier as well. For each $r_0$, let $r_1(r_0)$ and $p(r_0)$ denote the value and minimiser of the optimisation problem

$$\min_{p \in [0,1]} \{ r_1 \mid \text{both } \langle r_0, r_1 \rangle \pm \langle p, p - 1 \rangle \in \mathbb{G}_{T-1} \}.$$

We also refer to $\langle r_0, r_1(r_0) \rangle \pm \langle p(r_0), p(r_0) - 1 \rangle$ as the *rear* $(-)$ *and front* $(+)$ *contact points*. For $r_0 = 0$ we find $r_1(0) = T$, with witness $p(0) = 0$ and rear/front contact points $\langle 0, T + 1 \rangle$ and $\langle 0, T - 1 \rangle$, and for $r_0 = T$ we find $r_1(T) = 0$ with witness $p(T) = 1$ and rear/front contact points $\langle T - 1, 0 \rangle$ and $\langle T + 1, 0 \rangle$. It remains to consider the intermediate trajectory of $r_1(r_0)$ as $r_0$ runs from 0 to $T$. Initially at $r_0 = 0$ the rear contact point lies on the edge of $\mathbb{G}_{T-1}$ entering vertex $i = 0$ of $\mathbb{G}_{T-1}$, while the front contact point lies on the edge emanating from that same vertex. So if we increase $r_0$ slightly, the contact points will slide along their respective lines. By Lemma 11 (supplementary material), $r_1(r_0)$ will trace along a straight line as a result. Once we increase $r_0$ enough, both the rear and front contact point will hit the vertex at the end of their edges simultaneously (a fortunate fact that greatly simplifies our analysis), as shown in Lemma 12 (supplementary material). The contact points then transition to tracing the next pair of edges of $\mathbb{G}_{T-1}$. At this point $r_0$ the slope of $r_1(r_0)$ changes, and we have discovered a vertex of $\mathbb{G}_T$.

Given that at each such transition $\langle r_0, r_1(r_0) \rangle$ is the midpoint between both contact points, this implies that all midpoints between successive vertices of $\mathbb{G}_{T-1}$ are vertices of $\mathbb{G}_T$. And in addition, there are the two boundary vertices $\langle 0, T \rangle$ and $\langle T, 0 \rangle$. $\square$

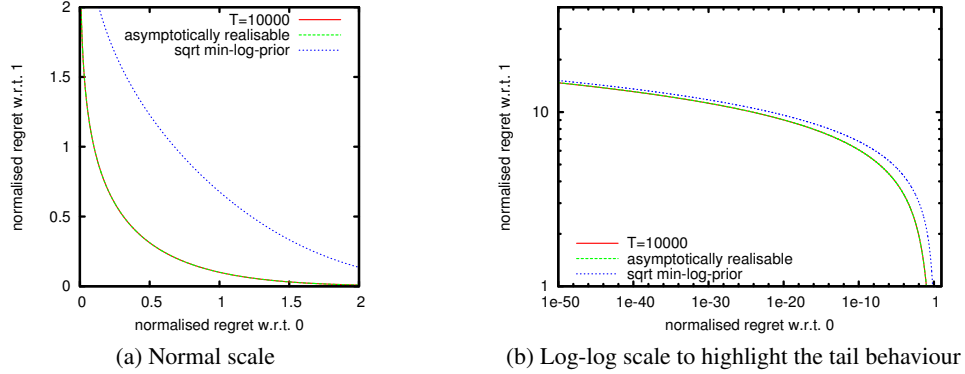

|  |  |
| :---: | :---: |
| (a) Normal scale | (b) Log-log scale to highlight the tail behaviour |

Figure 2: Pareto frontier of $\mathbb{G}$, the asymptotically realisable trade-off rates. There is no noticeable difference with the normalised regret trade-off profile $\mathbb{G}_T / \sqrt{T}$ for $T = 10000$. We also graph the curve $\left\langle \sqrt{-\ln(q)}, \sqrt{-\ln(1-q)} \right\rangle$ for all $q \in [0, 1]$.

## 3.1 The optimal strategy and random walks

In this section we describe how to follow the optimal strategy. First suppose we desire to witness a $T$-realisable trade-off that happens to be a vertex of $\mathbb{G}_T$, say vertex $i$ at $\langle f_T(i), f_T(T-i) \rangle$. With $T$ rounds remaining and in state $i$, the strategy predicts with $p_T(i)$. Then the outcome $x \in \{0, 1\}$ is revealed. If $x = 0$, we need to witness in the remaining $T-1$ rounds the trade-off $\langle f_T(i), f_T(T - i) \rangle - \langle p_T(i), p_T(i) + 1 \rangle = \langle f_{T-1}(i-1), f_{T-1}(T-1) \rangle$, which is vertex $i-1$ of $\mathbb{G}_{T-1}$. So the strategy transition to state $i-1$. Similarly upon $x = 1$ we update our internal state to $i$. If the state ever either exceeds the number of rounds remaining or goes negative we simply clamp it.

Second, if we desire to witness a $T$-realisable trade-off that is a convex combination of successive vertices, we simply follow the mixture strategy as constructed in Lemma 2. Third, if we desire to witness a sub-optimal element of $\mathbb{G}_T$, we may follow any strategy that witnesses a Pareto optimal dominating trade-off.

The probability $p$ issued by the algorithm is sometimes used to randomly sample a "hard prediction" from $\{0, 1\}$. The expression $|p - x|$ then denotes the expected loss, which equals the probability of making a mistake. We present, following [3], a random-walk based method to sample a 1 with probability $p_T(i)$. Our random walk starts in state $\langle T, i \rangle$. In each round it transitions from state $\langle T, i \rangle$ to either state $\langle T - 1, i \rangle$ or state $\langle T - 1, i - 1 \rangle$ with equal probability. It is stopped when the state $\langle T, i \rangle$ becomes extreme in the sense that $i \in \{0, T\}$. Note that this process always terminates. Then the probability that this process is stopped with $i = T$ equals $p_T(i)$. In our case of absolute loss, evaluating $p_T(i)$ and performing the random walk both take $T$ units of time. The random walks considered in [3] for $K \geq 2$ experts still take $T$ steps, whereas direct evaluation of the optimal strategy scales rather badly with $K$.

## 4 The asymptotic regret rate trade-off profile

In the previous section we obtained for each time horizon $T$ a combinatorial characterisation of the set $\mathbb{G}_T$ of $T$-realisable trade-offs. In this section we show that properly normalised Pareto frontiers for increasing $T$ are better and better approximations of a certain intrinsic smooth limit curve. We obtain a formula for this curve, and use it to study the question of realisability for large $T$.

**Definition 7.** Let us define the set $\mathbb{G}$ of *asymptotically realisable* regret rate trade-offs by

$$\mathbb{G} \ := \ \lim_{T \to \infty} \frac{\mathbb{G}_T}{\sqrt{T}}.$$

Despite the disappearance of the horizon $T$ from the notation, the set $\mathbb{G}$ still captures the trade-offs that can be achieved *with prior knowledge of $T$*. Each achievable regret rate trade-off $\langle \rho_0, \rho_1 \rangle \in \mathbb{G}$

may be witnessed by a different strategy for each $T$. This is fine for our intended interpretation of $\sqrt{T}\,\mathbb{G}$ as a proxy for $\mathbb{G}_T$. We briefly mention horizon-free algorithms at the end of this section.

The literature [2] suggests that, for some constant $c$, $\langle\sqrt{-c\ln(q)},\sqrt{-c\ln(1-q)}\rangle$ should be asymptotically realisable for each $q\in[0,1]$. We indeed confirm this below, and determine the optimal constant to be $c=1$. We then discuss the philosophical implications of the quality of this bound.

We now come to our second main result, the characterisation of the asymptotically realisable trade-off rates. The Pareto frontier is graphed in Figure 2 both on normal axes for comparison to Figure 1a, and on a log-log scale to show its tails. Note the remarkable quality of approximation to $\mathbb{G}_T/\sqrt{T}$.

**Theorem 8.** *The Pareto frontier of the set $\mathbb{G}$ of asymptotically realisable trade-offs is the curve*

$$\langle f(u), f(-u)\rangle \qquad for\ u\in\mathbb{R}, \qquad where \qquad f(u) := u\operatorname{erf}(\sqrt{2}u) + \frac{e^{-2u^2}}{\sqrt{2\pi}} + u,$$

*and $\operatorname{erf}(u) = \frac{2}{\sqrt{\pi}}\int_0^u e^{-v^2}\,\mathrm{d}v$ is the error function. Moreover, the optimal strategy converges to*

$$p(u) = \frac{1-\operatorname{erf}\left(\sqrt{2}u\right)}{2}.$$

*Proof.* We calculate the limit of normalised Pareto frontiers at vertex $i = T/2 + u\sqrt{T}$, and obtain

$$
\begin{aligned}
\lim_{T\to\infty}\frac{f_T\left(T/2+u\sqrt{T}\right)}{\sqrt{T}} &= \lim_{T\to\infty}\frac{1}{\sqrt{T}}\sum_{j=0}^{T/2+u\sqrt{T}}j2^{j-T}\binom{T-j-1}{T/2-u\sqrt{T}-1}\\
&= \lim_{T\to\infty}\frac{1}{\sqrt{T}}\int_0^{T/2+u\sqrt{T}}j2^{j-T}\binom{T-j-1}{T/2-u\sqrt{T}-1}\,\mathrm{d}j\\
&= \lim_{T\to\infty}\int_{-\sqrt{T}/2}^{u}(u-v)2^{(u-v)\sqrt{T}-T}\binom{T-(u-v)\sqrt{T}-1}{T/2-u\sqrt{T}-1}\sqrt{T}\,\mathrm{d}v\\
&= \int_{-\infty}^{u}(u-v)\lim_{T\to\infty}2^{(u-v)\sqrt{T}-T}\binom{T-(u-v)\sqrt{T}-1}{T/2-u\sqrt{T}-1}\sqrt{T}\,\mathrm{d}v\\
&= \int_{-\infty}^{u}(u-v)\frac{e^{-\frac{1}{2}(u+v)^2}}{\sqrt{2\pi}}\,\mathrm{d}v \quad = \ u\operatorname{erf}(\sqrt{2}u)+\frac{e^{-2u^2}}{\sqrt{2\pi}}+u
\end{aligned}
$$

In the first step we replace the sum by an integral. We can do this as the summand is continuous in $j$, and the approximation error is multiplied by $2^{-T}$ and hence goes to 0 with $T$. In the second step we perform the variable substitution $v = u - j/\sqrt{T}$. We then exchange limit and integral, subsequently evaluate the limit, and in the final step we evaluate the integral.

To obtain the optimal strategy, we observe the following relation between the slope of the Pareto curve and the optimal strategy for each horizon $T$. Let $g$ and $h$ denote the Pareto curves at times $T$ and $T+1$ as a function of $r_0$. The optimal strategy $p$ for $T+1$ at $r_0$ satisfied the system of equations

$$
\begin{aligned}
h(r_0)+p-1 &= g(u+p)\\
h(r_0)-p+1 &= g(u-p)
\end{aligned}
$$

to which the solution satisfies

$$1-\frac{1}{p} = \frac{g(r_0+p)-g(r_0-p)}{2p} \approx \frac{\mathrm{d}g(r_0)}{\mathrm{d}r_0}, \qquad so\ that \qquad p \approx \frac{1}{1-\frac{\mathrm{d}g(r_0)}{\mathrm{d}r_0}}.$$

Since slope is invariant under normalisation, this relation between slope and optimal strategy becomes exact as $T$ tends to infinity, and we find

$$p(u) = \frac{1}{1+\frac{\mathrm{d}f(r_0(u))}{\mathrm{d}r_0(u)}} = \frac{1}{1+\frac{f'(u)}{f'(-u)}} = \frac{1-\operatorname{erf}\left(\sqrt{2}u\right)}{2}.$$

We believe this last argument is more insightful than a direct evaluation of the limit. $\qquad\square$

## 4.1 Square root of min log prior

Results for Hedge suggest — modulo a daunting tuning problem — that a trade-off featuring square root negative log prior akin to (2) should be realisable. We first show that this is indeed the case, we then determine the optimal leading constant and we finally discuss its sub-optimality.

**Theorem 9.** *The parametric curve* $\left\langle \sqrt{-c\ln(q)}, \sqrt{-c\ln(1-q)} \right\rangle$ *for* $q \in [0,1]$ *is contained in* $\mathbb{G}$ *(i.e. asymptotically realisable) iff* $c \geq 1$.

*Proof.* By Theorem 8, the frontier of $\mathbb{G}$ is of the form $\langle f(u), f(-u) \rangle$. Our argument revolves around the tails (extreme $u$) of $\mathbb{G}$. For large $u \gg 0$, we find that $f(u) \approx 2u$. For small $u \ll 0$, we find that $f(u) \approx \frac{e^{-2u^2}}{4\sqrt{2\pi}u^2}$. This is obtained by a 3rd order Taylor series expansion around $u = -\infty$. We need to go to 3rd order since all prior orders evaluate to 0. The additive approximation error is of order $e^{-2u^2}u^{-4}$, which is negligible. So for large $r_0 \gg 0$, the least realisable $r_1$ is approximately

$$r_1 \approx \frac{e^{-\frac{r_0^2}{2} - 2\ln r_0}}{\sqrt{2\pi}}. \tag{3}$$

With the candidate relation $r_0 = \sqrt{-c\ln(q)}$ and $r_1 = \sqrt{-c\ln(1-q)}$, still for large $r_0 \gg 0$ so that $q$ is small and $-\ln(1-q) \approx q$, we would instead find least realisable $r_1$ approximately equal to

$$r_1 \approx \sqrt{c}\,e^{-\frac{r_0^2}{2c}}. \tag{4}$$

The candidate tail (4) must be at least the actual tail (3) for all large $r_0$. The minimal $c$ for which this holds is $c = 1$. The graphs of Figure 2 illustrate this tail behaviour for $c = 1$, and at the same time verify that there are no violations for moderate $u$. □

Even though the sqrt-min-log-prior trade-off is realisable, we see that its tail (4) exceeds the actual tail (3) by the factor $r_0^2\sqrt{2\pi}$, which gets progressively worse with the extremity of the tail $r_0$. Figure 2a shows that its behaviour for moderate $\langle r_0, r_1 \rangle$ is also not brilliant. For example it gives us a symmetric bound of $\sqrt{\ln 2} \approx 0.833$, whereas $f(0) = 1/\sqrt{2\pi} \approx 0.399$ is optimal.

For certain log loss games, each Pareto regret trade-off is witnessed uniquely by the Bayesian mixture of expert predictions w.r.t. a certain non-uniform prior and vice versa (not shown). In this sense the Bayesian method is the ideal answer to data compression/investment/gambling. Be that as it may, we conclude that the world of absolute loss is not information theory: simply putting a prior is *not* the definitive answer to non-uniform guarantees. It is a useful intuition that leads to the convenient sqrt-min-log-prior bounds. We hope that our results contribute to obtaining tighter bounds that remain manageable.

## 4.2 The asymptotic algorithm

The previous theorem immediately suggests an approximate algorithm for finite horizon $T$. To approximately witness $\langle r_0, r_1 \rangle$, find the value of $u$ for which $\sqrt{T}\langle f(u), f(-u) \rangle$ is closest to it. Then play $p(u)$. This will not guarantee $\langle r_0, r_1 \rangle$ exactly, but intuitively it will be close. We leave analysing this idea to the journal version. Conversely, by taking the limit of the game protocol, which involves the absolute loss function, we might obtain an interesting protocol and "asymptotic" loss function[2], for which $u$ is the natural state, $p(u)$ is the optimal strategy, and $u$ is updated in a certain way. Investigating such questions will probably lead to interesting insights, for example horizon-free strategies that maintain $R_T^k/\sqrt{T} \leq \rho_k$ for all $T$ simultaneously. Again this will be pursued for the journal version.

# 5 Extension

## 5.1 Beyond absolute loss

In this section we consider the general setting with $K = 2$ expert, that we still refer to as $0$ and $1$. Here the learner plays $p \in [0, 1]$ which is now interpreted as the weight allocated to expert $1$, the adversary chooses a loss vector $\boldsymbol{\ell} = \langle \ell_0, \ell_1 \rangle \in [0, 1]^2$, and the learner incurs *dot loss* given by $(1 - p)\ell_0 + p\ell_1$. The regrets are now redefined as follows

$$R_T^k := \sum_{t=1}^T p_t \ell_t^1 + (1 - p_t)\ell_t^0 - \sum_{t=1}^T \ell_t^k \qquad \text{for each expert } k \in \{0, 1\}.$$

**Theorem 10.** *The $T$-realisable trade-offs for absolute loss and $K = 2$ expert dot loss coincide.*

*Proof.* By induction on $T$. The loss is irrelevant in the base case $T = 0$. For $T > 0$, a trade-off $\langle r_0, r_1 \rangle$ is $T$-realisable for dot loss if

$$\exists p \in [0, 1] \forall \boldsymbol{\ell} \in [0, 1]^2 : \langle r_0 + p\ell_1 + (1 - p)\ell_0 - \ell_0, r_1 + p\ell_1 + (1 - p)\ell_0 - \ell_1 \rangle \in \mathbb{G}_{T-1}$$

that is if

$$\exists p \in [0, 1] \forall \delta \in [-1, 1] : \langle r_0 - p\delta, r_1 + (1 - p)\delta \rangle \in \mathbb{G}_{T-1}.$$

We recover the absolute loss case by restricting $\delta$ to $\{-1, 1\}$. These requirements are equivalent since $\mathbb{G}_T$ is convex by Lemma 2. □

## 5.2 More than $2$ experts

In the general experts problem we compete with $K$ instead of $2$ experts. We now argue that an algorithm guaranteeing $R_T^k \leq \sqrt{cT \ln K}$ w.r.t. each expert $k$ can be obtained. The intuitive approach, combining the $K$ experts in a balanced binary tree of two-expert predictors, does not achieve this goal: each internal node contributes the optimal symmetric regret of $\sqrt{T/(2\pi)}$. This accumulates to $R_T^k \leq \ln K \sqrt{cT}$, where the log sits outside the square root.

Counter-intuitively, the *maximally unbalanced* binary tree does result in a $\sqrt{\ln K}$ factor when the internal nodes are properly skewed. At each level we combine $K$ experts one-vs-all, permitting large regret w.r.t. the first expert but tiny regret w.r.t. the recursive combination of the remaining $K - 1$ experts. The argument can be found in Appendix A.1. The same argument shows that, for any prior $q$ on $k = 1, 2, \ldots$, combining the expert with the smallest prior with the recursive combination of the rest guarantees regret $\sqrt{-cT \ln q(k)}$ w.r.t. each expert $k$.

# 6 Conclusion

We studied asymmetric regret guarantees for the fundamental online learning setting of the absolute loss game. We obtained exactly the achievable skewed regret guarantees, and the corresponding optimal algorithm. We then studied the profile in the limit of large $T$. We conclude that the expected $\sqrt{T}\langle \sqrt{-\ln(q)}, \sqrt{-\ln(1 - q)} \rangle$ trade-off is achievable for any prior probability $q \in [0, 1]$, but that it is not tight. We then showed how our results transfer from absolute loss to general linear losses, and to more than two experts.

Major next steps are to determine the optimal trade-offs for $K > 2$ experts, to replace our traditional $\sqrt{T}$ budget by modern variants $\sqrt{L_T^k}$ [15], $\sqrt{\frac{L_T^k(T - L_T^k)}{T}}$ [16], $\sqrt{\text{Var}_T^{\max}}$ [17], $\sqrt{D_\infty}$ [18], $\Delta_T$ [19] etc. and to find the Pareto frontier for horizon-free strategies maintaining $R_T^k \leq \rho_k \sqrt{T}$ at any $T$.

**Acknowledgements**

This work benefited substantially from discussions with Peter Grünwald.

## Footnotes

[1]One could define the regret $R_T^k$ for all static reference probabilities $k \in [0, 1]$, but as the loss is minimised by either $k = 0$ or $k = 1$, we immediately restrict to only comparing against these two.

[2] We have seen an instance of this before. When the Hedge algorithm with learning rate $\eta$ plays weights $\boldsymbol{w}$ and faces loss vector $\boldsymbol{\ell}$, its *dot loss* is given by $\boldsymbol{w}^T\boldsymbol{\ell}$. Now consider the same loss vector handed out in identical pieces $\boldsymbol{\ell}/n$ over the course of $n$ trials, during which the weights $\boldsymbol{w}$ update as usual. In the limit of $n \to \infty$, the resulting loss becomes the *mix loss* $-\frac{1}{\eta}\ln\sum_k w(k)e^{-\eta\ell_k}$.

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
