[Supplementary Material · appendix.pdf]

# A    Proofs

**Lemma 11** (Stick Tracing). *Fix straight lines $L = \{\langle r_0, r_1\rangle \mid ar_0 + br_1 + c = 0\}$ and $\Lambda = \{\langle r_0, r_1\rangle \mid \alpha r_0 + \beta r_1 + \gamma = 0\}$. The points $\langle r_0, r_1\rangle$ such that there is a $p$ for which the point $\langle r_0, r_1\rangle + \langle p, p-1\rangle$ lies on the line $L$ and the point $\langle r_0, r_1\rangle - \langle p, p-1\rangle$ lies on the line $\Lambda$ form a straight line.*

*Proof.* The points $\langle r_0, r_1\rangle$ in question satisfy for some $p$

$$a(r_0 + p) + b(r_1 + p - 1) + c = 0,$$
$$\alpha(r_0 - p) + \beta(r_1 - p + 1) + \gamma = 0.$$

Eliminating $p$, we find that the solution set equals

$$\big(a(\alpha + \beta) + \alpha(a+b)\big)r_0 + \big(b(\alpha + \beta) + \beta(a+b)\big)r_1 + (a+b)(\beta + \gamma) + (\alpha + \beta)(c - b) = 0,$$

which is a straight line as required.  $\square$

**Lemma 12.** *We use the notation of Theorem 6. The system of linear equations*

$$\langle r_0, r_1\rangle - \langle p, p-1\rangle = \langle f_{T-1}(i-1), f_{T-1}(T-i)\rangle$$
$$\langle r_0, r_1\rangle + \langle p, p-1\rangle = \langle f_{T-1}(i), f_{T-1}(T-1-i)\rangle$$

*has unique solution*

$$r_0 = f_T(i), \quad r_1 = f_T(T-i) \quad and \quad p = p_T(i).$$

*Proof.* The solution set of the system can be rewritten to

$$\langle r_0, r_1\rangle = \frac{\langle f_{T-1}(i), f_{T-1}(T-1-i)\rangle + \langle f_{T-1}(i-1), f_{T-1}(T-i)\rangle}{2}$$

$$\langle p, 1-p\rangle = \frac{\langle f_{T-1}(i) - f_{T-1}(i-1), f_{T-1}(T-i) - f_{T-1}(T-1-i)\rangle}{2}$$

Notice that the system is over-constrained, so we are essentially checking that it involves a redundant constraint. It remains to verify that the proposed solution fits. We do this for $r_0$ and $p$, the cases for $r_1$ and $1 - p$ follow by symmetry when exchanging $i$ and $T - i$.

To see that $r_0 = f_T(i)$, we rewrite

$$f_{T-1}(i-1) + f_{T-1}(i) = \sum_{j=0}^{i-1} j 2^{j-T+1} \binom{T-j-2}{T-i-1} + \sum_{j=0}^{i} j 2^{j-T+1} \binom{T-j-2}{T-i-2} =$$

$$2\sum_{j=0}^{i} j 2^{j-T} \left( \binom{T-j-2}{T-i-1} + \binom{T-j-2}{T-i-2}\right) = 2\sum_{j=0}^{i} j 2^{j-T}\left(\binom{T-j-1}{T-i-1}\right) = 2f_T(i)$$

The case for $p = p_T(i)$ holds by definition.  $\square$

## A.1    More than 2 experts

We now show how to achieve the bound $R_T^k \leq \sqrt{-cT \ln q(k)}$ for an arbitrary prior $q$. Our construction is a recursive combination of asymmetric binary strategies. The crux is to combine the experts one-vs-all, with the expert with lowest prior vs the rest. Note that we may always assume that the number of experts $K$ is finite (in fact $K \leq \sqrt{T}$), as the bound trivially holds for each expert $k$ with $-c \ln q(k) \geq \sqrt{T}$.

Fix a prior $q(k)$ on $k = 1, \ldots, K$ ordered by increasing probability. In this section we for simplicity work from the $\langle \sqrt{-\ln p}, \sqrt{-\ln(1-p)}\rangle$ trade-off (this is achievable, see Section 4.1). We combine the expert with smallest prior with the recursive combination of the others. We employ the combination parametrised by $p = q(1)^{-c}$ for some fixed universal constant $c$ determined below. We claim that this combination guarantees $R_T^k \leq \sqrt{-cT \ln q(k)}$ for each $k$. The proof is by induction. The

recursive combination that combines expert 1 vs the rest, guarantees regret w.r.t. expert 1 bounded by

$$\sqrt{-cT \log q(1)}$$

and that w.r.t. each expert $k > 1$ by

$$\sqrt{-T \log(1 - q(1)^c)} + \sqrt{-cT \log \frac{q(k)}{1 - q(1)}}$$

It remains to show that we can choose $c$ such that

$$\sqrt{-T \log(1 - q(1)^c)} + \sqrt{-cT \log \frac{q(k)}{1 - q(1)}} \leq \sqrt{-cT \log q(k)}$$

that is

$$\sqrt{-\log(1 - q(1)^c)} \leq \sqrt{-c \log q(k)} - \sqrt{-c \log \frac{q(k)}{1 - q(1)}}.$$

As the square root is concave the right-hand side increases with $q(k) \geq q(1)$, so we need to show

$$\sqrt{-\log(1 - q(1)^c)} \leq \sqrt{-c \log q(1)} - \sqrt{-c \log \frac{q(1)}{1 - q(1)}}.$$

It is rather complicated to determine analytically the least $c$ that achieves this for all $q \leq 1/3$, or get a good bound. However, a straightforward numerical plot shows that $c = 2.51202$ is sufficient.