[Reviews · NeurIPS 2013]

Submitted by Assigned_Reviewer_3

This paper concerns regret against individual actions in the full-information online prediction setting. Instead of trying to bound the regret uniformly for all experts (actions), the work gives characterization of what "regret profiles" are achievable for the most basic case of binary prediction with absolute loss. The author(s) call a profile < r0,r1 > T-realizable if there exists an algorithm that guarantees regret r0 and r1 against constant predictions 0 and 1, respectively, given any (non-oblivious) adversary. An example for a trivial T-realizable profile is < 0,T >.

The paper exactly (even constants) characterizes the Pareto-front of all T-realizable profiles for any T in the basic case of two actions and absolute loss. Then they use the result to prove an asymptotic bound. The paper also provides algorithms that achieve the realizable profiles.

The paper is well written, the problem is well motivated, the proofs seem sound. The problem of satisfying non-uniform regret bounds is an interesting new line of research. This paper takes the first steps in this new direction.



Some comments:

In Definition 1, the paper reveals that the adversary model considered is the non-oblivious (adaptive) one. I would prefer if it was mentioned more explicitly and at an earlier point in the paper. Especially because Lemma 3 does not go through for oblivious adversaries.

I found the second part of Theorem 6 confusing. First I thought it was a full description of the algorithm that achieves the vertices of the Pareto-front. Only later I realized that it is just some fact about what the algorithm does at three particular time steps. Maybe say this explicitly or just leave out this part.

How do the new bounds relate to the old uniform regret bound of \sqrt{T/2 \ln K}? I would have liked to see some discussion about this, especially since the above bound is asymptotically optimal.

The algorithms that prove the upper bounds seem to be highly inefficient computationally. It would be nice to have algorithms that are more efficient and have regret bounds close to the optimal ones.
Summary: This work gives optimal bounds for regret-profiles against each constant action in online learning. The paper is technically sound and it opens a new line of research in regret analysis.

Submitted by Assigned_Reviewer_4

The authors give an exact characterization of the tradeoffs achievable between regret-vs-expert-0 and regret-vs-expert-1 in a two expert problem with known horizon T. They provide an algorithm for achieving any tradeoff on the Pareto frontier, with the point corresponding to an equal bound on each expert’s regret recovering the standard definition. They also consider the asymptotic behavior of this tradeoff curve.

The primary contributions of the paper are Theorems 6 and 8. These are nice results, particularly Thm 6 which provides a tight minimax characterization. It would be nice if more motivation for the key function f_T(i) was provided, it seems to appear magically. I suspect there are some Rademacher random variables and binomial probabilities underneath this somewhere, with Thm 8 coming from something like a normal approximation to the binomial. Unfortunately, these results are also quite limited in that they only apply to the K=2 expert case. Sec 4.1 points out that for K=2 the the sqrt(min log prior) frontier is weaker, but this does not seem surprising, since this bound comes from an algorithm that is not minimax optimal.

Theorem 10 is also important, as it shows Thms 6 and 8 apply to the general experts problem in 2 dimensions, rather than just to the special case of the absolute loss. The proof suggests, in fact, that one could simply re-state the paper in terms of the more general problem; I think this would be a better presentation, but at a minimum Thm 10 should be moved earlier in the paper.

I’m not clear on what Sec 5.2 is trying to accomplish. It seems to indicate that standard sqrt(T log K) bounds can be recovered from the K=2 algorithm presented here. But the authors need to be clear about this, and argue why it is useful or interesting. After all, we already have much simpler algorithms that achieve this bound.

The main drawback of the paper is that it does not address the K > 2 case, which is clearly the one of the most practical interest.

Summary: The authors fully characterize the Pareto frontier for prediction with expert advice, but unfortunately only in the rather limited case of 2 experts.

Submitted by Assigned_Reviewer_5

This paper studies regret guarantees in prediction with expert advice that are simultaneously achievable against each individual expert. The problem is essentially solved for the case of two constant binary experts under the absolute loss (or dot loss). In particular, the finite-time and asymptotical region of achievable pairs are computed. Connections to random playout and bounds in terms of log prior weights are discussed.

This is a solid and well written paper that explores a new direction in the theory of prediction with expert advice. The results are precise and mathematically elegant, although a bit narrow in scope. I do not expect a significant impact on the NIPS community, although there is a latent potential here that futher research might be able to fully express.

The techniques seem to strongly rely on the linearity of the loss, it would be good to say something about, say, convex and Lipschitz losses.

The minimax algorithm for absolute loss and fixed horizon has been first proposed in: N. Cesa-Bianchi et al., How to use expert advice. Journal of the ACM, 44(3):427-485, 1997. Please fix the reference.

===============================

I have read the authors' rebuttal.
Summary: A solid and elegant paper with somewhat narrow results. I would accept it, although I expect it to be interesting only for a small fraction of the NIPS community
Author Feedback

Author rebuttal: Dear reviewers and program chairs,

Thank you for your time invested in making NIPS a high-quality conference.

We are happy with the quality and thoroughness of the reviews. We are also grateful for the generous "high impact" score by Reviewer 3, and we indeed hope that our paper will open a new line of research in algorithm design and regret analysis, harvesting the latent potential envisioned by Reviewer 5.

Thank Reviewer 4 for pointing out a naive misjudgement on our behalf, for which we apologise: When we wrote "absolute loss" we meant the vanilla regret notion where regret is measured compared to all static actions. As the loss is linear, we may immediately restrict to actions 0 and 1. In this view "absolute loss" implies 2 experts. This is explained in Section 2. However, thanks to your comments we realised that of course absolute loss is used more widely, and it does not carry our intended connotation. We agree that this must absolutely be clarified in the final paper, to prevent any "oversell", which only results in subsequent reader frustration.

We share with Reviewer 4 the desire to crack the K>2 case. Unfortunately however, this is by no means a trivial extension of K=2. To appreciate this, observe that even the symmetric minimax regret analysis for K=3 experts and fixed T is hard, because the optimal algorithm has to be reined in, to prevent it from placing negative weight on experts. This does not happen in the K=2 case. These boundary effects at some game states then percolate to (and interact in) all encompassing longer games. We burned through a series of conjectures about K>2 by performing numeric calculations of the minimax strategy for small T, and a characterisation by means of formulas remains highly elusive. It will probably be the case that only asymptotic results can be obtained.

We would argue that our exact solution, even though only for the case K=2, is an important theoretical contribution by itself and a solid starting point for K>2.

Finally, we would like to reassure Reviewer 3 that the "Moreover" part of Theorem 6 indeed describes the optimal learner strategy at all vertices. The left two formulas apply at the boundary vertices, and the right main formula applies to all internal vertices. After learner follows the specified strategy at vertex i with T rounds remaining, the resulting trade-off to be realised in the remaining T-1 rounds is again a vertex (the outcome determines whether it is i or i-1), to which the formulas apply again, and so on.